# Recombinant Spidroin Microgel as the Base of Cell-Engineered Constructs Mediates Liver Regeneration in Rats

**DOI:** 10.3390/polym14153179

**Published:** 2022-08-04

**Authors:** Murat Shagidulin, Nina Onishchenko, Anastasiia Grechina, Alla Nikolskaya, Mikhail Krasheninnikov, Aleksey Lyundup, Elena Volkova, Natalia Mogeiko, Artem Venediktov, Gennadii Piavchenko, Lubov Davydova, Alla Ramonova, Vladimir Bogush, Sergey Gautier

**Affiliations:** 1Federal State Budgetary Institution “Shumakov National Medical Research Center of Transplantology and Artificial Organs” of the Ministry of Health of the Russian Federation, 123182 Moscow, Russia; 2Federal State Autonomous Educational Institution of Higher Education, “I.M. Sechenov First Moscow State Medical University” of the Ministry of Health of the Russian Federation (Sechenov University), 119435 Moscow, Russia; 3Research and Education Resourse Center, Peoples Friendship University of Russia (RUDN University), 117198 Moscow, Russia; 4National Research Center “Kurchatov Institute”, 123182 Moscow, Russia; 5Bioengineering Department, Faculty of Biology, M.V. Lomonosov Moscow State University, 119234 Moscow, Russia

**Keywords:** chronic liver failure, matrix, liver cells, mmsc bm, cell-engineered constructs

## Abstract

*Aim*: In this study, we seek to check if recombinant spidroin rS1/9 is applicable for cell-engineering construct development. Novel technologies of cell and tissue engineering are relevant for chronic liver failure management. Liver regeneration may represent one of the possible treatment options if a cell-engineered construct (CEC) is used. Nowadays, one can see the continuous study of various matrices to create an appropriate CEC. *Materials and Methods*: We have adhered allogenic liver cells and multipotent mesenchymal bone marrow stem cells (MMSC BM) to a microgel with recombinant spidroin rS1/9. Then we have studied the developed implantable CEC in a rat model (*n* = 80) of chronic liver failure achieved by prolonged poisoning with carbon tetrachloride. *Results:* Our results demonstrate that the CECs change the values of biochemical tests and morphological parameters in chronic liver failure in rats. *Conclusion:* We consider there to be a positive effect from the microgel-based CECs with recombinant spidroin rS1/9 in the treatment of chronic liver failure.

## 1. Introduction

Clinical transplantation suffers from a constantly increasing lack of donor organs [1,2,3]. Nowadays, meeting this challenge requires an efficient treatment and even replacement of vital organs and tissues to compensate for their functions. Our study seeks to develop new, elaborate and relevant ways of such a replacement.

The efficacy and safety of liver cell transplantation has already been demonstrated in patients, though its therapeutic benefit remains partial or transient, and most of the patients require further orthotopic liver transplantation [4]. The problem is probably due to a limited number of hepatocytes that enter the portal vein with no embolic action, as well as a low engraftment rate of the transplanted hepatocytes in the parenchyma of recipient livers [5]. In addition, the survival time of isolated liver cells is short.

Hybrid systems for extracorporeal perfusion (e.g., bioartificial liver) also demonstrate some disadvantages and limitations calling for the development of intracorporeal devices with appropriate conditions for longer survival and higher efficiency of donor liver cells. Liver tissue engineering proposes the creation of functioning tissues in vitro and in vivo. The approach promises to solve the task by supporting liver functions in vivo for a long time period [5].

Both tissue engineering [6,7,8] and bioartificial organ construction [9,10] involve cell implantation onto carriers or into microcapsules to enable cell regeneration treatment. Various experimental and clinical studies have shown that almost all new cells die in a relatively short period after transplantation with no carrier due to the lack of normal differentiation and proliferation. Several components are, therefore, crucial: cells producing a functioning extracellular matrix; appropriate carriers for cell transplantation; and bioactive molecules (cytokines, growth factors) that exhibit a biostimulating action. The components can be administered as cell suspensions by short (2 to 4 h) extracorporeal perfusions. Thus, one may expect longer donor cell functioning if there is an intracorporeal device with a biocompatible three-dimensional carrier imitating the organ structure or with a cell-engineered construct (CEC).

The carrier should provide a framework facilitating cell attachment. It should also degrade in vivo over time. It is important to get the degradation rate similar to the rate of extracellular matrix deposition and cell proliferation [11]. Finally, such a carrier should establish a nutrient-rich environment to maintain cell viability. Tissue-like scaffold materials provide a growth medium for cell communication and functional maintenance [12]. For example, gelatin methacryloyl (GelMA), sodium alginate and polyethylene glycol (PEG) are widely used as common carrier materials owing to their high biocompatibility and mechanical performance for liver cell proliferation and differentiation [12]. To get an ideal CEC, we should consider not only these properties but also: ability to stimulate cell proliferation and differentiation; ability to avoid any local inflammatory reaction; ability to avoid any toxic and immunogenic effects; absence of oncogenic effects or genotoxicity; absence of infectious risks; maintenance of functional properties during all the expected service life; and availability for routine sterilization with no loss of its medical and technical properties.

Recombinant spider web proteins (also called recombinant spidroins or RSs) meet all of these requirements for implantable CECs. RSs are analogues of dragline silk proteins in spider webs. These substances were developed by the National Research Center’s “Kurchatov Institute”. RSs fulfil all the necessary physical, chemical and biological criteria owing to their high elasticity, toughness, resistance to high and low temperatures, amphiphilicity and self-assembly as well as to their high biocompatibility and absence of toxic or allergenic action. In addition, they do not transmit any infections to humans [13,14]. Compared to collagen and various synthetic polymers, RSs induce a weaker immune response when implanted [15].

In this study, we use RS rS1/9 the gene of wich has already been designed, synthesized and cloned (in *Saccharomices cerevisiae* yeast cells) [16]. The sequence of forming elements is similar to the natural sequence of spidroin 1 for the backbone filament in spider webs of *Nephila clavipes*. The protein rS1/9 consists of 9 identical monomers. Any monomer contains 4 primary chains [12]. Such a chain includes a poly-A-block of 5–8 A residues; it is enriched also by GGX tripeptides (X = L, Y, Q). The structure of rS1/9 is thereby periodic while its molecular weight reaches 94 kDa. Some previous data assume that the poly-A-blocks in spidroins are responsible for crystalls formation to improve the filament strength [17]. Poly-A-crystals are surrounded by amorphous G-containing regions providing the elasticity of spider silk [18]. This protein (like all the products deriving from it) exposes pretty high pl H value (10.3). As a result, it acquires positive charges over the entire range of physiological pH values. Therefore, one may remark an adhesion to cells whose surfaces are usually negative-charged.

Using self-assembling properties of RSs, we have developed various samples of 2D and 3D structures for carriers, such as non-woven materials (obtained by electrospinning), transparent films, sponge-like 3D matrices, hydrogels, microgels and microcapsules. The matrices are remarkable for the spontaneous formation of nano- and micropores inside their membranes. They thereby participate in proper nutrition and oxygenation of administered cells [16]. Toxicological studies have revealed that hydrogels and microgels are harmless agents with neither irritating nor sensitizing action in animals [19]. In addition, numerous tests of 2D and 3D RSs-deriving products (rS1/9 and rS2/12; the latter one is a recombinant analogue of spidroin 2 in spider web scaffold filaments of *Nephila madagascariensis*) in vitro and in vivo have demonstrated high levels of biocompatibility for all of them, as well as correlation between the rate of their bioresorption and the rate of new tissue formation [20,21,22,23,24].

We have chosen RS rS1/9 as a microgel to be the material for our cell-engineered construct with liver cells. Available literature sources do not contain data about studies of RSs microgels in production of CECs for the compensation and treatment of chronic liver failure (CLF). Therefore, the aim of our study has been the technological development of methods for the preparation of CECs with liver cells (LC) and mesenchymal multipotent stromal bone marrow cells (MMSC BM), which are immobilized on a microgel that is a derivative of recombinant rSs1/9 spidroin.

The hydrogel structure can be formed from spidroin solutions by physical and/or chemical crosslinking. During physical crosslinking, the forming of a stable hydrogel is induced by increasing the concentration of RS solutions, as well as by acidification and/or by changing the ionic composition of the solution or by shear forces. The conformation rearrangement occurs in spidroins upon their transformation into hydrogel. It leads to forming of β-sheet nanocrystalline structures [25]. Hydrogels can be mechanically fragmented and fractionated according to particle size. In our study, microgels were obtained from rS1/9 with a particle size in the range of 50–300 µm. This structure can be fractionated and conveniently delivered to certain body sites using a syringe. It is known that the surface relief of such particles is complex, promoting adsorption and proliferation of cells [26]. Due to the mild conditions of the assembly, it is relatively easy to add various substances and particles to the gels thus bringing them to living cells and obtaining 3D cell constructs for tissue engineering [27]. In laboratory animals, bioengineered rS1/9-derived microgels have exerted the function of a scaffold for cells but have also influenced the immune response; they have shown pro-regenerative properties as well [28,29].

## 2. Materials and Methods

### 2.1. Microgel: Description and Preparation

We used microgel (based on recombinant spidroin rS1/9) to create a cell-engineered liver construct. rS1/9 has a molecular mass of 94 kDa, pI = 10.3 and contact angle 56.6°.

We performed yeast biomass production as well as isolation and purification of recombinant spidroin rS1/9 in accordance with the previously published protocol [30]. The protein after chromatographic purification was, dialyzed against deionized water, frozen at −70 °C and freeze-dried.

Hydrogel samples were produced from a 3% solution of highly purified RS (electrophoretic purity 96 ± 1%) as described previously [28]. After dialysis, in order to initiate gelation, MgCl_2_ was added to the protein solution at a final concentration of 2 mM; then, this resulting solution was incubated at 30 °C for 24 h before hydrogel formation. We obtained microgels crushing hydrogel physically using a 300 μm sieve. The resulting suspension constituted a heterogeneous mixture of microgel particles ranging from 50 to 300 µm in diameter, with a predominance of 100 to 300 μm particles. The microgel samples form a transparent suspension including gel-like particles of classic hydrogel. The microgel’s water content is 97 ± 0.2%, while rS1/9 electrophoretic purity reaches 96 ± 1%. 48 ± 3% of rS1/9-based microgel decays because of hydroxyl radicals of Fenton’s reagent until the 3rd week (unpublished data). High pI level (10.3) of spidroin rS1/9 makes it be obligatorily positive at any pH value in the in vivo range, and this is also true for rS1/9-based microgel.

### 2.2. Microgel Surface Morphology Analysis

Preparation of microgel for scanning electron microscopy (SEM) was performed in accordance with the standard protocol. Briefly, microgel particles were fixed overnight using 2.5% glutaraldehyde in 0.1 M cacodylate buffer at +4 °C. The samples were then washed three times in 0.1 M cacodylate buffer at pH = 7.2 for 5 min, followed by dehydration in a series of ethanol solutions with increasing concentrations and acetone (Chemmed, Moscow, Russia). After critical point drying using Hitachi critical point dryer HCP-2 (Hitachi Ltd., Tokyo, Japan), microgel particles were metallized with a 20-nm-thick platinum layer using Ion Coater IB3 (Eiko Engineering Co., Hitachinaka, Japan). The resulting samples were analyzed with Camscan S2 microscope (Cambridge Instruments, London, UK) at 10-nm resolution and operating voltage −20 kV. Images were obtained using Micro Captures (SMA, Moscow, Russia).

### 2.3. Animals, CLF Model

Experimental studies included male rats (Wistar) at an age of 6–8 months and weighing 230–250 g. The animals were kept in a vivarium at a temperature of 18–20 °C on a mixed diet with free access to water. Experiments with animals took place from 9 am until 7 pm at room temperature (t = 22–24 °C) so that these periods covered possible daily fluctuations in the mitotic activity of liver cells. Experiments and all manipulations with animals were carried out in accordance with the rules adopted by the European Convention for the Protection of Vertebrate Animals used for research and other scientific purposes (European Convention for the Protection of Vertebrate Animals Used for Experimental and other Scientific Purposes (ETS 123), Strasbourg, 1986).

The model of CLF was achieved by prolonged poisoning of Wistar rats (*n* = 80) by carbon tetrachloride according to our modified scheme for 42 days [31].

### 2.4. Obtaining LC and MMSC BM

The donors of allogeneic LC and MMSC BM were male Wistar rats aged 5–6 months, weighing 150–230 g.

The work on the isolation of cells and their cultivation was conducted in accordance with the general principles of the implementation of cultural studies. The preparation of the MMSC BM culture was carried out according to the generally accepted method [31,32].

The LC culture was prepared according to the procedure [31]. The colonization (immobilization) of the matrix with cellular material was fulfilled according to the well-known method on the 7th day after the end of the CLF modeling [32].

First, we performed preliminary co-cultivation of isolated LCs (2.5–4.0 × 10^6^ cells/cm^3^) and MMSC BM (0.5–0.8 × 10^6^ cells/cm^3^) for 3 days in a William’s E (REF 22551-022) growth medium using the ratio of these cells: LC + MMSC BM = 5:1 [33]. An additional 150 µL volume of microgel suspensions based on RS rS1/9 was injected into the cell culture for their adhesion. Cytodex-3 (Sigma-Aldrich Product Number: C3275, CAS Number: 88895-19-6, MDL number: MFCD00130902) was added (volume of 150 µL) in order to ensure the transplanted cellular material in the CEC for long periods after their implantation in the liver. The resulting suspension’s volume (containing LC, MMSC BM, rS1/9, and Cytodex-3) was adjusted up to 1 mL by William’s E (REF 22551-022) growth medium. The resulting volume was fractionally injected into the liver parenchyma.

The viability of isolated, cultured, and co-cultivated allogeneic LCs and MMSC BM before implantation was determined by trypan blue staining (Q. Chen 2007, P. Godoy 2013).

### 2.5. Group Characterization and Study Design

On the 7th day after the end of the poisoning by CCl_4_, the rats that survived after CLF modeling (*n* = 60) were divided into 2 groups. The effectiveness of CLF correction was evaluated after implantation of the prepared CECs into the parenchyma of the damaged liver: 1—control group (CLF + 1 mL of saline, *n* = 20); 2—experimental group (*n* = 40) where CECs (based on RS rS1/9 and containing allogeneic LC and MMSC BM = 5:1 and Cytodex-3) were injected into the parenchyma of the damaged liver by chipping in a volume of 1 mL.

We controlled adequacy of the developed models (as well as effectiveness of CEC-dependent morphofunctional correction in the liver) by assessment of mortality, survival, morphological and morphometric liver properties. Biochemical blood parameters permitted assessing the functional efficiency of CEC application. After CLF modeling either with or without treatment, surviving animals were euthanized on day 90 by intraperitoneal administration of sodium thiopental at a dosage causing respiratory arrest. We then realized morphological studies of the slides with liver sections. The experiments did not involve any immunosuppression.

### 2.6. Parameters to Study

We used MATLAB software to transform histological images into images of 3D surfaces. Thus, we identified the following types of structures: “oxyphilic cytoplasm of hepatocyte”, “basophilic nuclei of hepatocyte”, “connective tissue proper”, “intercellular substance” and “empty spaces (vascular lumen, adipocytes)”. The original image was converted to a gray scale. A 3D surface was built (taking into account the added constants for each gray-scale image), the models and colors were specific for each image [34]. Rats were anesthetized by ether to take blood samples for biochemical studies by notching the tail tip. We assessed liver function (e.g., ALT, AST, ALP) by Reflotron™ biochemical analyzer (Roche, Switzerland) using Reflotron test strips (REF: AST–107 45 120, ALT–107 45 138, ALP–116 22 773). Morphological study steps discerned the state of regenerative processes in the liver, both in the CEC implantation zones and out of them. Liver sections were stained with hematoxylin and eosin, by Mallory, or by Masson. We used a Leica DM 6000 B microscope (Wetzlar, Germany) and a Leica LTDCH 9435 camera (Wetzlar, Germany).

Immunohistochemical reactions were carried out using monoclonal antibodies against proliferating cell nucleus marker, PCNA (DakoCytomation, Hamburg, Germany). Deparaffinized sections were pretreated in a citrate buffer for 10 min. The liver sections were then incubated with the antibody for 30 min at room temperature. The labeled streptavidin-biotin method (LSAB; DakoCytomation, Germany) detection system, based on the biotin-streptavidin complex and peroxidase, has been used to reveal the reaction (sequential 10-min incubations with biotinylated link antibody and peroxidase-labelled streptavidin). Diaminobenzidine was used as a visualization substrate. After the reaction, the sections were additionally stained with hematoxylin and eosin. All the necessary controls were carried out.

Staining for type I collagen was carried out using pAb rabbit anti-collagen I aff. purified (No. 29047Y, Monosan, Uden, The Netherlands). Fluorescent staining anti-Human HNF-4a (Lifetechnology, Carlsbad, CA, USA) with detection of goat anti rabbit IG, alexa fluor REF: 417700, Lot: 74002428A.

The obtained results were processed by statistical package Biostat; we employed the t-test to check for significant differences, taking the Bonferonni correction into account. These differences were considered as the significant for *p* < 0.05 (Statistical package recommended by WHO, EpiInfo 5.0).

The survival of animals was calculated according to Kaplan–Meier by using the statistical package of the program Statistica for Windows, v.12 for either animals during the first year after CLF modeling (control) or after CLF treatment by implanted CECs.

## 3. Results

The surface structure of microgel is analyzed by SEM. The microgel constitutes microparticles whose average size ranges from 100 to 300 μm while the surface has a complex relief. The surface elements include nanostructures (diameter of 100 to 300 nm) and microstructures (10 to 30 μm) as well as micropores of 2–70 μm in diameter (Figure 1). These nanostructures facilitate interactions with cells; the great number of pores including nanopores [29] and the surface positive charge perform the same function.

We have conducted a comparative evaluation of the efficiency for implantable rS1/9-based CECs containing allogeneic LC and MMSC BM (5:1) for the treatment of CLF to assess survival terms and dynamics as well as restoration of biochemical and morphological parameters in recipient livers in 90 days.

The viability of MMSC BM in primary culture reaches 94 ± 2%. The cell suspension prepared from a donor liver contains ~95 ÷ 98% hepatocytes and ~5 ÷ 2% non-parenchymal cells, while cell viability reaches 76 ± 4%. The cells have not been separated into parenchymal and non-parenchymal cells.

Having obtained the primary cell culture, we have carried out a pre-cultivation to eliminate stress damage of cells during isolation and to activate their functions. MMSC BM were cultivated for 7 days, and then co-cultivated with LC introduced into the culture during the next 3 days (Figure 2). The cell co-cultivation has been performed in our lab not only in a stationary mode, but also in a rotational mode (1.5–2.0 rotations/min) to simulate the conditions of mass transfer provided by blood circulation, because only these conditions provide long-term viability and proliferative activity of LC during their co-cultivation with MMSC BM (Figure 2F).

The CLF model (within 42 days) caused a lethality of 25%. In the control group (Gr. 1) after the CLF modeling, 5 rats (from the remaining 20 rats) died additionally during the whole experimental period (90 days). Thus, in the control group, the high rat lethality during 90 days of observation confirms the adequacy of the chosen model of serious CLF. In the experimental group (Gr. 2), there was no lethality among rats for the same period (42 + 90 days). Survival of rats at the stage of poisoning and conducting of treatment is shown in Figure 3.

The liver injury after modeling CLF has been accompanied by a sharp increase in the levels of liver enzymes, namely ALT, AST and alkaline phosphatase (Figure 4A–C). In 7 days after completion of poisoning, the level of AST increased by more than 4.5 times, the level of ALT increased by more than 3 times and the level of alkaline phosphatase increased by almost 5 times.

The study has shown that the recovery of biochemical parameters (ALT, AST and alkaline phosphatase) to normal values can be observed in the experimental group (Gr. 2) during the first 28–30 days remaining within the normal range, while in the control group these parameters have remained elevated for 90 days (Figure 4); moreover, during this period there has been additional lethality in the control group (Figure 3B).

Histological examination after the toxic damage of the liver (Figure 5) in the CLF modeling has revealed necrosis of hepatocytes and their fatty degeneration, which manifested in the appearance of a large number of hepatocytes with degenerating nuclei and intranuclear lipid inclusions, whereas the number of binuclear hepatocytes decreased significantly. In addition, we have detected: a loss of glycogen in the cytoplasm of the hepatocytes; pronounced polymorphism of the parenchymal cells; focal hemorrhages in the liver parenchyma; fatty and macrofocal vacuolar dystrophy of the liver cells; early growth of connective tissue; and development of liver fibrosis (-portal and porto–central fibrosis) together with significant inflammatory infiltration in the fibrosis areas and formation of false lobules (Figure 5).

In rat livers of the control group 90 days after CLF modeling, there has been a subtotal reorganization of the liver histoarchitectonics with the replacement of normal parenchyma by false lobules. Sclerotic changes (fibrosis) were manifested in the formation of collagen fibers along the portal tracts and also by formation of porto–portal and porto–central septa together with formation of false lobules. There is a pronounced protein degeneration of hepatocytes, as well as a focal necrosis of hepatocytes, sclerosis and fibrosis of the liver parenchyma; moreover, there is expansion and a plethora of sinusoids and a plethora of central veins in the parenchyma. In addition, rare lymphoid cell infiltration, proliferation of histioblasts and histiocytes and progression of liver fibrosis were revealed (Figure 5B–D).

The assessment of the liver morphological state at 90 days in the experimental Gr. 2 has revealed pronounced positive dynamics of the hepatic parenchyma restoration. Histological examination of the liver in this group has shown that the liver structures in the areas that are not adjacent to the CECs were completely restored (Figure 5) with no practical difference from the normal liver. We have observed a regression of fatty degeneration in hepatocytes and its transition to a small droplet form (more favorable and reversible). Furthermore, in the restoration of the hepatic lobule structure, intact hepatocytes have emerged without signs of dystrophy around veins.

Thus, the total architectonics of the liver after the CLF modeling and implantation of CECs was recovered. False lobules were not found, while their formation was evident in the control group (Gr. 1) (Figure 5E–G).

In CLF treatment, it is important to use cells as a part of CECs that can function for the long term, establishing their restorative and regulatory actions in the damaged liver tissue of the recipient. This is why we have studied the state of implanted CECs and cells (LCs and MMSC BM) immobilized on their matrix—RS rS1/9.

The obtained results of histological examination of CECs (Figure 5) indicate that allogeneic LCs within the CECs retain their viability and proliferative activity without signs of rejection for a long time (90 days) in spite of the absence of immunosuppressive therapy. We have observed that the shape, and the nuclei-to-cytoplasm ratio in hepatocytes return, then, to the normal values, and, as before the CLF modeling, fibrous connective tissue capsule emerges around them (Figure 5H–J).

Figure 5K,L represents the liver tissue with PCNA-positive proliferating hepatocytes (brown) among the neighboring cells additionally stained with hematoxylin and eosin on the 90th day of experiment. The identification of tissue region with CEC was due to the localization of Cytodex-3.

## 4. Discussion

After transplantation of donor liver cells or activation of auxiliary liver systems in CLF, the analysis of results revealed the advantages and disadvantages of these methods. It made it possible to introduce a concept how to improve their effectiveness [1,2,3].

The approach tends to provide conditions for a longer life of donor cells and long-term bioregulatory action in damaged liver. To achieve this, one should attach the cellular material to a biocompatible and biodegradable 3D matrix of cell-engineered constructs (CECs) before their implantation into the recipient’s body.

Nowadays, a number of studies have already been carried out with CECs containing liver cells (LC) and multipotent mesenchymal stem cells of bone marrow (MMSC BM) [27,28]. To optimize the conditions for the formation of CECs and maintain optimal biological properties [16,17,18,19,20], we decided to include recombinant spidroin in their composition. We found that the best option comprises the following steps. First, to improve adhesion one needs to pre-cultivate MMSC BM for 7 days and, then, to co-cultivate these cells for 3 days together with freshly isolated donor LC before planting them on RSs-matrices.

In this study, we investigated the possibility of using such a matrix as a microgel derived from recombinant spidroin rS1/9. The microgel formed a basis of CECs containing allogeneic LC and MMSC BM to modulate regeneration processes in the damaged liver tissue in an optimal way. In a large number of experiments, RS rS1/9 proved that its regenerative capabilities are very impressive due to high biocompatibility, positive charge, slow bioresorption in vivo and the ability to maintain adhesion and proliferation of various cell types. The microgel is a suspension of 100–300 µm sized particles of a physically crushed hydrogel. It obtains a highly porous structure, soft texture and positive charge, becoming the basis to form CECs and to treat liver damage. The microgel of such CECs (based exclusively on protein) cannot only be a framework for maintaining cells but also a substrate for their nutrition. In addition, due to the small size of the microgel particles, the suspension of such CECs can be injected directly into the damaged organ by a syringe injection.

In this work, the high efficiency of CECs with allogeneic LC and MMSC BM (ratio of 5:1) was remarkable. There was no mortality among rats in the experimental group within 90 days after modeling of serious chronic liver failure. For the control group (rats did not receive treatment by CECs), mortality rate reached 25% for the same period (5 rats). Comparative histological study where morphological state of the liver was checked in recipients on day 90 showed that the experimental group had almost the same liver architectonics as a normal liver in areas not adjacent to the CECs. This is evidenced by the restoration of the damaged structure, by a significant decrease of new connective tissue area, a decrease in the number of hepatocytes with adipose degeneration, and a decrease in nuclear degeneration of liver tissue cells.

A biochemical assessment of changes in the activity of liver enzymes (ALT, AST, alkaline phosphatase) showed that all enzymes recovered to normal levels within the first 28–30 days and remained within normal limits up to 90 days after the administration of CECs. At the same time, these parameters were still elevated in the control group until the end of the experiment.

We performed the analysis of comparative studies (lack of mortality, rapid return into normal ranges for enzymes, and restoration of the damaged liver structure in the experimental group) for day 90 after CLF modeling. We consider that CECs with microgel (RS rS1/9 and allogeneic LC and MMSC BM in a ratio of 5:1) contributes to a more rapid functional recovery of the damaged liver in CLF modeling. After the implantation, CECs based on RS rS1/9 together with allogeneic LC and MMSC BM can become crucial for de novo formation of new liver regeneration centers and provide their prolonged action. A prolonged bioregulatory effect of the implanted CECs enables restorative regeneration of the damaged liver as well as the correction and treatment of liver failure.

An additional role of the obtained results is the fact that during the experiment no external factors (such as drugs, biologically active molecules or nutrients) were introduced, and no immunosuppression was used.

Thus, the obtained data confirm that the microgel based on RS rS1/9 contributes to the maintenance of the vital activity of adhesive cells in the CECs implanted in the liver, and, therefore, the microgel is advisable to use for the formation of implantable CECs for the correction and treatment of CLF.

## 5. Conclusions

The results of biochemical and histological studies in animals with chronic liver failure modeling reveal expressed functional activity of CECs containing the microgel based on RS rS1/9 and cellular material (allogeneic LC and MMSC BM in a ratio of 5:1).

We found that the administration of CECs, implanted in the liver parenchyma, leads to the normalization of biochemical indices of liver function already by 30 days after implantation (unlike the control group) and to the restoration of morphological state of the liver within 90 days.

The histological and artificial intelligence analysis of the liver state after implantation of CECs in liver parenchyma at the experimental toxic hepatitis showed that the structural changes in the liver tissue are well-distinguished and notable in histological specimens and remarkably classified, being identified by various colors. As a result, the new methods allow one to see the different tissue areas with different tinctorial properties [35,36] and confirm the activation of regenerative processes in liver when using the CECs.

It has also been shown that the cells included in the CECs not only lived and developed up to 90 days, but also initiated the positive structural changes in the damaged liver parenchyma. These facts indicate the suitability of the microgel based on rS1/9 as a scaffold for production of CECs and also safety of their use for a long-term implantation in the organism. The high efficiency of using the CECs, containing the microgel based on rS1/9 and also high biocompatibility of this matrix with cells and tissues, the absence of immunogenic properties and the relatively low cost of their production indicate the possibility and expediency using the CECs containing the microgel based on rS1/9 as a new biotechnological method of treating severe chronic diseases in clinical practice.

## Figures and Tables

**Figure 1 polymers-14-03179-f001:**
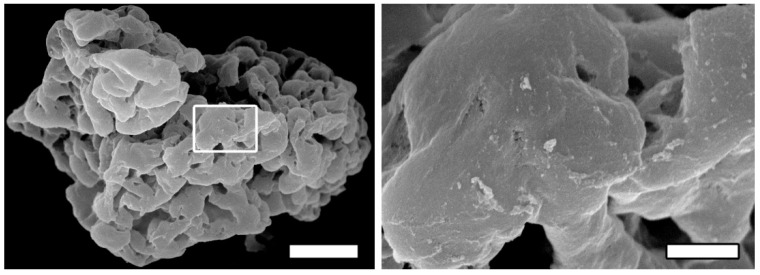
Structure of rS1/9 microgel. The left panel shows a SEM image of rS1/9 microgel. Scale bar 30 μm. The right panel represents the region in the white frame on the left. Scale bar 3 μm.

**Figure 2 polymers-14-03179-f002:**
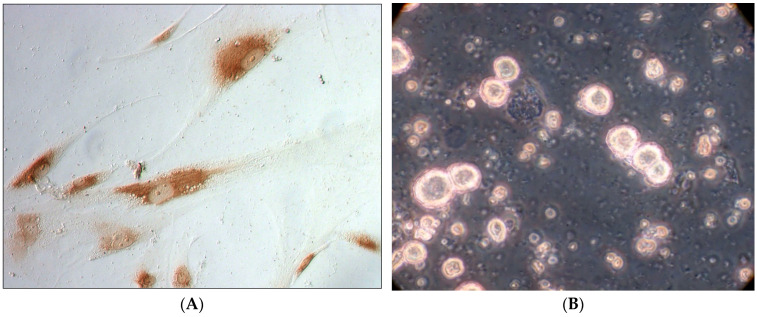
Culture of rat MMSC BM after 7 days of cultivation. (**A**)—staining for type I collagen, ×400; (**B**)—hepatocytes, phase contrast ×400; (**C**)—liver cells, fluorescent stain by murine anti-Human HNF-4a (lifetechnology); (**D**)—MMSC BM and liver cells after 3 days of co-cultivation, phase contrast, ×200; (**E**)—culture of rat MMSC BM, ×400; (**F**)—co-cultivation of LC and MMSC BM on rS1/9, phase contrast, ×400.

**Figure 3 polymers-14-03179-f003:**
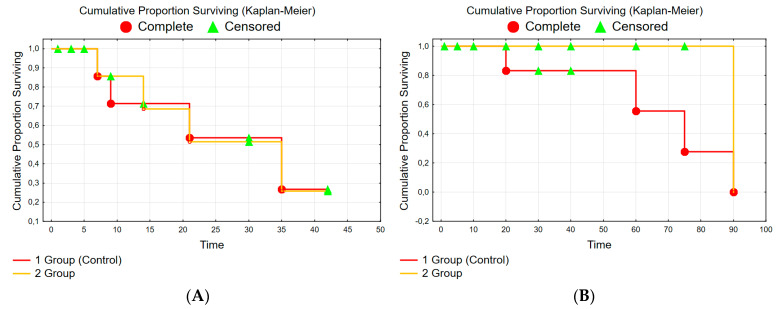
Survival of experimental rats (Kaplan–Meier) (**A**)—modeling of CLF Gehan’s Wilcoxon Test (Spreadsheet1) WW = −1000 Sum = 670.00 Var = 76.32 Test statistic = −0.037655 *p* = 0.96996 and (**B**)—treatment of CLF: Gehan’s Wilcoxon Test (Spreadsheet1) WW = −11.00, Sum = 238.00, Var = 62.632, Test statistic = −1.32676, *p* = 0.18459.

**Figure 4 polymers-14-03179-f004:**
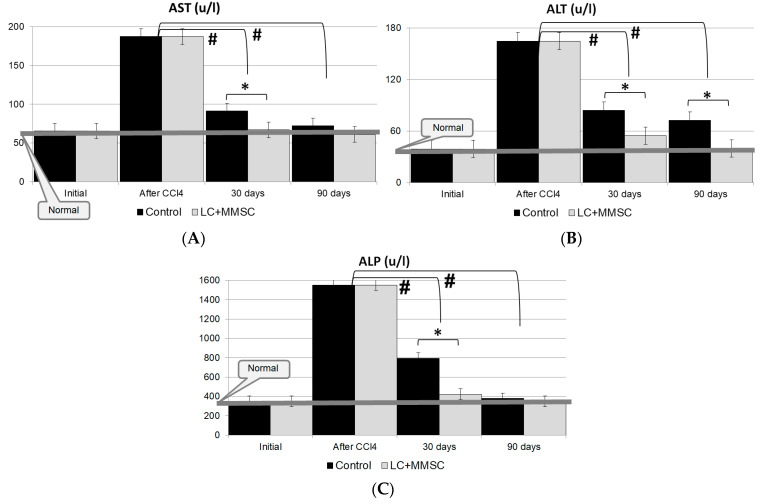
Dynamics of enzymatic parameters: (**A**)—AST, (**B**)—ALT; (**C**)—ALP in the rat blood serum after CLF modeling and CECs implantation. Gr. 1—control (infusion of saline); Gr. 2—implantation of CECs based on RS rS1/9 with allogeneic LC:MMSC BM = 5:1. *—The difference is significant compared with the level of enzymes in the control (Group 1); *p* < 0.05. #—The difference is significant compared with the level of enzymes right away after CLF modelling; *p* < 0.05.

**Figure 5 polymers-14-03179-f005:**
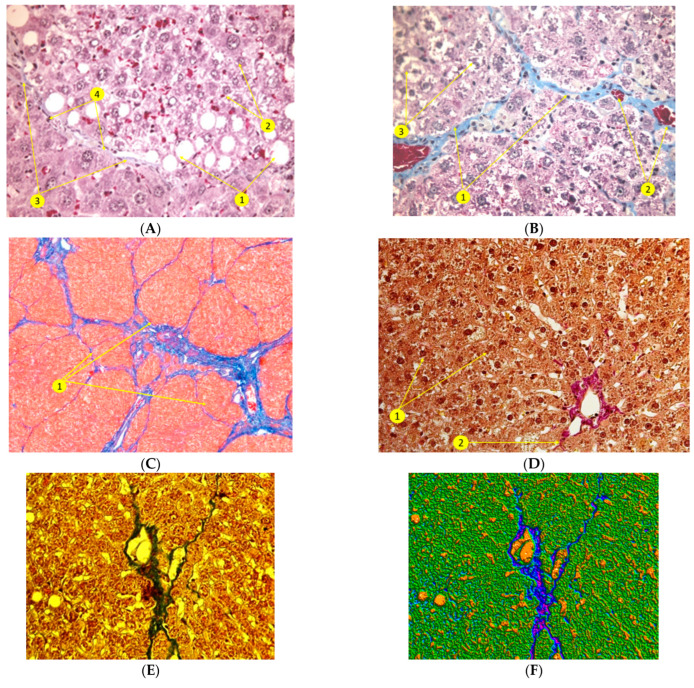
(**A**)—rat liver tissue after chronic liver failure modeling: Initial formation of fibrous tissue, protein and fatty degeneration of hepatocytes (1), necrotic hepatocytes (2), early growth of connective tissue (3), proliferation of histioblasts and macrophages into fibrous septa (4). Staining by Masson, ×100. Histological slides of the rat liver 90 days after completion of CLF modeling (**B**–**D**): (**B**)—porto–portal fibrosis (1). Stasis of blood in the veins (2). Staining by Masson, ×400. (**C**)—false lobules. Connective tissue staining by Mallory, × 100. (**D**)—Protein degeneration of hepatocytes (1). Periportal fibrosis (2). Staining by Van Gieson, ×400. Histological structures of the liver (Gr. 2) 90 days after the CLF modeling and implantation of CECs (**E**–**G**): (**E**)—The central part of the lobule with remaining cell structures. Trichrome staining by Masson, ×200. (**F**,**G**)—pathologically modified hepatocytes with fatty dystrophy and fibrous connective tissue capsule and surrounding healthy hepatocytes. Hepatic parenchyma is pseudo-stained as green, fibrous connective tissue capsule is blue with some parts of high density (violet), empty spaces in orange. CECs, containing allogeneic LC and MMSC BM (group 2), which were implanted in liver after 90 days of CLF modeling (**H**–**J**): (**H**)—transplanted living hepatocytes as a part of the CEC. Hematoxylin-eosin staining. ×400; (**I**,**J**)—CEC is in the center of the image, with the normal hepatocytes surrounded by fibrous connective tissue envelope; CEC is implanted into the hepatic parenchyma. After artificial intelligence analysis we have classified the following structures: fibrous connective tissue envelope and cell nuclei are blue, normal cytoplasm of hepatocytes is cyan, hypertrophic cytoplasm of hepatocytes is green, lumen is yellow and hyperchromic nuclei are violet. (**K**,**L**)—Liver parenchyma and fibrous connective tissue. PCNA-positive cells (1)—light brown with dark brown nuclei, hepatocytes (2)—oxyphilic cytoplasm and basophilic nuclei, fibrous connective tissue (3)—interlobular hepatic venule of the triad (4)—PCNA IHC staining under standard protocols with additional H&E staining ×100.

## Data Availability

The data presented in this study are available on request from the corresponding author.

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
