# Peer review of "Recombinant Spidroin Microgel as the Base of Cell-Engineered Constructs Mediates Liver Regeneration in Rats"

_polymers, 2022, doi:10.3390/polym14153179_

Round 1
Reviewer 1 Report
The manuscript presented by the authors describes a new alternative approach to liver transplantation in cases of CLF. The novelty lies in the use of a protein animal derivative, the spidroin, which, in the form of a microgel, allows primary cells to be delivered by injection directly into the organ to be treated. Moreover, by proceeding in advance with an in-vitro culture phase of the cells on these microgels, the actual effectiveness of the treatment seems to increase.
The research work is indeed interesting, but unfortunately, the manuscript in its present state of being is in no shape to be published. First of all, a major revision of English, both in grammar and syntax, is necessary. Some sentences are long and not easy to understand. Furthermore, spelling errors are found throughout the manuscript. Overall, one has the feeling that the work has been completed in every section, from polymer preparation through cell culture to transplantation, but reading on there is a noticeable lack of characterisation data that would support the work.
The introduction section is chaotic. The limits and disadvantages of organ transplantation and of the available alternative approaches (cell transplantation and BAL) are presented, but there is no clear distinction between the latter two. BAL devices contain cells to ideally supplant the liver function of a patient awaiting transplantation, cell transplantation aims to treat the patient directly. This difference is not presented and should be. From this, CECs would be proposed as an attractive and improved alternative to existing and/or investigational procedures.
The materials and methods section is not well organised and each paragraph should be improved in terms of lists of characterizations/analysis carried out to achieve the results presented further on in the manuscript.
The biomaterial section should be more detailed. Adding spidroin characterisation data would increase the value of the manuscript (permeability/porousness data, contact angle, etc.) as biocompatibility and cellular response in vitro and in vivo depend on them. An additional description must be added with respect to the physical state of the microgel. Is it a classic hydrogel (geltrex, matrigel, collagen) or a solid matrix/microcarrier wherein resuspends the cell culture, as implied by the expression 'colonisation (immobilisation) of the matrix with cellular material'?
As far as the cell section is concerned, again, the data must be implemented and better described, especially in the description of the cell culture performed prior to transplantation. For example, the sentence stating "an additional 150 uL of microgel suspension [...] was injected to the cell culture" leads to confusion. What is referred to as an injection? Is the gel inserted into a fixed cell culture system (matrix) or does it mean addition of gel to the system (hydrogel)?
Linked to previous comments, the results section should be improved. More data concerning microgels, cells and the resulting CECs should be added. There are no references in the materials and methods of a dynamic system used to maintain cell culture when it is presented in the results. What exactly does it consist of?
Although the post-transplantation data are encouraging, the section describing them needs to be reworded so that the data and observations are clearer on the first reading, as it has to be done for the discussion section.
Reviewer 2 Report
In the present study, the authors implanted cell-engineered constructs in the liver parenchyma of CCl4 intoxicated rats. The authors demonstrated that the microgel based on recombinant spidroin rS1/9 normalized biochemical indices of liver function and restored morphological state of the liver.
My comments
- In the abstract, the model used for chronic liver failure should be indicated
-Introduction needs to be summarized and to be more focused. For example, the first paragraph can be removed
-Some parts in introduction need citations such as properties of CECs
-Materials and methods as well as results should be separated into subheadings for better organization
-Indicate catalogue number for kits used for assessment of ALP, AST, ALT
-Figure 4, no need to mention the normal value of the parameters tested. Also, the exact value of p should be indicated
-Changes observed in histological studies should be indicated on figures by arrows, arrowheads, etc. Also figures from different experimental groups including normal should be combined in one figure for better presentation
-Figure 7, the first figure correct F to A
-Ki-67 and PCNA Immunostaining could be performed to ensure liver regeneration
Reviewer 3 Report
The Author’s data indicate that microgel, spidroin rS1/9 is able to grow hepatocytes and cell-engineered constructs can treat chronic liver failure. The spidroin rS1/9 holds onto hepatocytes for 90 days and is involved in maintenance of liver function. However, the presentation of the data and analysis is not well organized and inconclusive.
First, the abstract should represent the background, aim, method, results and conclusion. It is not well organized.
The main body needs to be re-organized. The figure legends were not separated from the content, it will be difficult read.
For the data presentation, a control group should be shown against treatment or experimental groups. For example, figure 2 needs a control group of rat MMSC BM alone, second panel will be liver cell and rat MMSC BM with liver cells. Then explain each panel and deduce any benefit or conclusion.
After 90 days, cell viability was up to 94%. Post-culture, the cells should be checked for their type, especially LCs need to be confirmed by hepatocytes markers.
Histological analysis of the liver 90 days after CLF modeling and implantation of CECs: How can one identify the spidroin rS1/9 mixed with cell in recipient rat liver? Used arrowheads or cell markers (if transfected protein marker) or sex mismatched marker (Y chromosome), etc. In the figure, donor derived cells (CECs) are very difficult to identify in the recipient liver (CLF).
Round 2
Reviewer 1 Report
Although the manuscript has been spruced up with the addition of new data and the basic idea remains very interesting indeed, unfortunately, I think it would be appropriate for the authors to take the time to rewrite the manuscript so that each section is well defined and of high quality.
In fact, there are still many passages in the manuscript that are not easy to read; the figure references do not always correspond to what is given in the text (fig 4 and fig 2 for example) as well as the introduction still seems too fragmentary and not entirely well described. Moreover, the description of the results should be enhanced and described with more accuracy.
Reviewer 2 Report
The authors addressed most of comments
In figure 5, number E in labeling the figures is missed
Round 3
Reviewer 1 Report
apart from some minor revisions in formatting and punctuation, the manuscript has considerably increased in quality and scientific rigour